# Opportunities to sustain a multi-country quality of care network: Lessons on the actions of four countries Bangladesh, Ethiopia, Malawi, and Uganda

**Seblewengel Lemma**[1]\*, **Callie Daniels-Howell**[2], **Asebe Amenu Tufa**[3], **Mithun Sarker**[4], **Kohenour Akter**[4], **Catherine Nakidde**[5], **Gloria Seruwagi**[5], **Albert Dube**[6], **Kondwani Mwandira**[6], **The QCN Evaluation Group**[¶], **Desalegn Bekele Taye**[7], **Mike English**[8], **Yusra Ribhi Shawar**[9], **Kasonde Mwaba**[2], **Nehla Djellouli**[2], **Tim Colbourn**[2], **Tanya Marchant**[1]

1 Department of Disease Control, London School of Hygiene & Tropical Medicine, based in Ethiopia, Addis Ababa, Ethiopia, 2 Institute for Global Health, University College London, London, United Kingdom, 3 Health System Directorate, Ethiopian Public Health Institute, Addis Ababa, Ethiopia, 4 Perinatal Care Project, Diabetic Association of Bangladesh, Dhaka, Bangladesh, 5 School of Public Health, Makerere University, Kampala, Uganda, 6 Parent and Child Health Initiative PACHI, Lilongwe, Malawi, 7 Ministry of Health, Health Service Quality Directorate, Addis Ababa, Ethiopia, 8 Centre for Tropical Medicine and Global Health, University of Oxford, Oxford, United Kingdom, 9 Bloomberg School of Public Health and Paul H. Nitze School of Advanced International Studies, John Hopkins University, Baltimore, Maryland, United States of America

¶ Membership of the QCN Evaluation Group is listed in the Acknowledgments.
\* Seblewengel.Abreham@lshtm.ac.uk

**Data Availability Statement:** All data is derived from qualitative interviews, most with stakeholders where only one individual holds a position, either

## Abstract

The Quality of Care Network (QCN) is a global initiative that was established in 2017 under the leadership of WHO in 11 low-and- middle income countries to improve maternal, newborn, and child health. The vision was that the Quality of Care Network would be embedded within member countries and continued beyond the initial implementation period: that the Network would be sustained. This paper investigated the experience of actions taken to sustain QCN in four Network countries (Bangladesh, Ethiopia, Malawi, and Uganda) and reports on lessons learned. Multiple iterative rounds of data collection were conducted through qualitative interviews with global and national stakeholders, and non-participatory observation of health facilities and meetings. A total of 241 interviews, 42 facility and four meeting observations were carried out. We conducted a thematic analysis of all data using a framework approach that defined six critical actions that can be taken to promote sustainability. The analysis revealed that these critical actions were present with varying degrees in each of the four countries. Although vulnerabilities were observed, there was good evidence to support that actions were taken to institutionalize the innovation within the health system, to motivate micro-level actors, plan opportunities for reflection and adaptation from the outset, and to support strong government ownership. Two actions were largely absent and weakened confidence in future sustainability: managing financial uncertainties and fostering community ownership. Evidence from four countries suggested that the QCN model would not be sustained in its original format, largely because of financial vulnerability and insufficient time to embed the innovation at the sub-national level. But especially the efforts made

within federal or state government, facilities, or NGOs. Every care has been taken to ensure anonymity of the data in the submitted manuscript but the authors from all 4 countries feel strongly that making data freely available would jeopardise the conditions of informed consent. Data is available upon reasonable request to point of contact at the London School of Hygiene and tropical Medicine: researchdatamanagement@lshtm.ac.uk.

**Funding:** This work was funded by the Medical Research Council (MRC) Health Systems Research Initiative 5th call via grant MR/S013466/1 to TC at UCL Institute for Global Health, United Kingdom, YS and JS at Johns Hopkins University, United States of America, KA and AK at Diabetic Association of Bangladesh Perinatal Care Project, Bangladesh, CM at Parent and Child Health Initiative, Malawi, GS at Makerere University School of Public Health, Uganda, and ME at University of Oxford, United Kingdom; and by the Bill & Melinda Gates foundation via grant INV-007644 to TM at LSHTM, United Kingdom. The funders had no role in study design, data collection and analysis, decision to publish, or preparation of the manuscript.

**Competing interests:** The authors have declared that no competing interests exist.

to institutionalize the innovation in existing systems meant that some characteristics of QCN may be carried forward within broader government quality improvement initiatives.

## Introduction

The Quality of Care Network (QCN) is a global initiative that was established in 2017, motivated by the slow progress of countries in reducing maternal and newborn mortality, especially from preventable causes [1]. Evidence on the lack of equitable access to high quality health services for mothers, newborn and children [2] prompted the publication of standards and guidelines that promote high quality care [3, 4]. Support for country-driven action plans for sustainable, high-quality care was recognised as a gap. Under the leadership of the World Health Organization (WHO), QCN was established to address that gap, with eleven participating Network countries namely Bangladesh, Côte d'Ivoire, Ethiopia, Ghana, India, Kenya, Malawi, Nigeria, Sierra Leone, Uganda, and the United Republic of Tanzania. In addition to these country governments and the WHO, QCN also encompassed implementing, technical and donor partner organisations. Together these countries and partners created a platform for learning to understand how to implement and sustain quality of care initiatives at national and sub-national levels [1]. This paper concludes the collection of papers to examine the performance of QCN, focusing on four Network countries: Bangladesh, Ethiopia, Malawi, and Uganda (S1 Text). Here, we focused on the sustainability of the Network after five years of development and implementation.

Despite its importance, the concept of sustainability is not yet well defined and there is inadequate effort to measure sustainability of innovations [5–7]. In this paper we take sustainability of health programs to mean the continuity of a program after the implementation phase [5]. It is important that this continuity be planned alongside program implementation in order for communities to reap the long term benefit of interventions [5, 8, 9]. Without planning for sustainability, externally funded innovations that do not have strong government ownership are likely to lose momentum and cease to function when the funding agency withdraws or stops its support [9–11].

In an attempt to understand and potentially pre-empt this, studies have tried to identify the factors affecting sustainability and scaleup [5, 7, 10–14]. Building from these, Wickremasinghe and colleagues refined and summarized six actions that a donor funded innovation can implement to promote sustainability. These actions are (1) planning opportunities for reflection and adaptation from the outset (to ensure that innovations are fit for purpose through continuous engagement with government, and relevant stakeholders); (2) supporting strong government ownership with a plan for a phased transition of responsibilities as external partners withdraw (to ensure government support for and commitment to current and future implementation success); (3) motivating micro-level actors (to ensure that the needs and gaps of local level actors are understood such that they are enabled to engage and implement the innovation. In this paper, micro-level actors are health care workers and the supporting team at the lower level of the health system); (4) institutionalizing the innovation within the health system (to ensure that implementation is embedded within existing systems to enhance ownership, efficiency and reduce duplication); (5) managing financial uncertainties (to ensure financial commitment from governments such that innovation costs are included in the government budget plan); and (6) fostering community ownership (to ensure that community groups, for example clients of the health service or community groups, have the opportunity to catalyse the

continuity of the innovation through advocacy and ensure accountability in the implementation of the innovation [10].

This paper investigated the experience of actions taken to sustain QCN in four Network countries (Bangladesh, Ethiopia, Malawi, and Uganda) and reports on lessons learned.

## Method

This analysis was part of the multi-country evaluation of QCN, the methods of which are reported in our common methods supplement for our QCN Evaluation collection of papers (S2 Text). Key aspects of the methods in relation to this paper are summarized here.

### Study setting

The study was conducted in four QCN countries, namely Bangladesh, Ethiopia, Malawi, and Uganda; the study was started in 2018 except in Ethiopia that joined the study in 2019. An overview of key country characteristics is provided in Table 1.

**Bangladesh.** Maternal, newborn and child health (MNCH) is a priority agenda for Bangladesh with a population of more than 165 million [15]. According to the national health, population, and nutrition sector plan for the year 2017–2022, the government of Bangladesh has striven to improve the health of mothers and newborns through making home delivery safe, improving access to and utilization of emergency obstetric services, and improving access to newborn and child health care at the lower level of the health system [16]. Since 2017, the government of Bangladesh with implementing partners launched the QCN; it currently has 28 learning districts out of 62 districts, where Quality Improvement (QI) activities have been implemented (Table 1).

**Ethiopia.** The second most-populous country in Africa, Ethiopia achieved its Millennium Development Goals (MDGs) for maternal and child health [17]. There have been a number of government-led initiatives that explicitly address quality improvement and most recently, the Ministry of Health (MOH) adopted the national maternal and newborn quality of care roadmap for the year 2017–2020 [18]. This roadmap closely aligns with QCN activities which have been implemented in 14 learning districts out of 770 districts [19].

**Malawi.** Malawi is less populous compared to the other case study countries [20] (Table 1). Following its success in achieving its MDG target for child health, the MOH in Malawi engaged in initiatives that aimed to improve the health of mothers and newborns. The

**Table 1. Demographic and mortality characteristics for the four case study countries.**

| Characteristics | Bangladesh | Ethiopia | Malawi | Uganda |
|---|---|---|---|---|
| Total population size (million)[1] | 166.3 | 117.9 | 19.6 | 47.1 |
| Total number of districts | 64 | 832 | 28 | 136 |
| Maternal Mortality Ratio per 100,000[2] | 173 | 401 | 349 | 336 |
| Under 5 Mortality Rate per1000[3] | 29.3 | 59 | 59.1 | 58.4 |
| Neonatal Mortality rate per 1000[4] | 17 | 33 | 19 | 19 |
| Date launched QCN | 2017 | 2017 | 2017 | 2017 |
| Number of QCN learning districts | 28 | 14 | 6 | 6 |
| Number of QCN learning facilities | 298 | 48 | 25 | 18 |

[1] Population size from World Bank 2021 https://data.worldbank.org/indicator [15, 17, 20, 22]

[2] Bangladesh, Ethiopia, and Malawi MMR estimates from World Bank 2017 [25, 26]; Uganda from UDHS2016 [27]

[3] Under 5 MR Bangladesh, Malawi and Uganda(global age -sex-specific fertility and mortality rate 2019) [28]; Ethiopia (Mini-DHS 2019) [29]

[4]NMR Ethiopia (Mini-DHS 2019), UNICEF DATA (2020) Bangladesh, Malawi and Uganda [25, 26, 29, 30]

country established the Quality Management Directorate (QMD) within the MOH to improve service quality, addressed quality of service in its Health Sector Strategic Plan (HSSP-II) and developed its National Quality Policy and Strategy [21]. The MOH along with its partners have been implementing QI interventions in six learning districts out of 28 total districts in the country.

**Uganda.**   Uganda with a population size of more than 47 million [22] is also striving to improve quality of health service provision to improve the health of mothers and newborns. Uganda's adoption of various components of quality in healthcare dates back to 1994 [23] initially driven by quality management interventions in HIV/AIDS, TB and malaria. In the recent past, the national standards, guidelines, and policies on maternal and newborn health (MNH) quality of care (QoC) as well as the health sector QI framework and health sector strategic plan 2015/16–2019/20 have been developed. The MOH has begun to implement QI interventions in six learning districts out of 111 total districts in the country [24].

## Design

The study employed a mixed method design. To explore the actions taken by the QCN actors that affect the potential for sustainability, a thematic analysis [31] of qualitative interview data and observations from the participating four countries and from interviews with global-level actors was conducted.

## Data collection

For the purpose of this analysis, two data sources were accessed across the four countries (Table 2), and described below.

**Semi-structured interviews.**   First, semi-structured qualitative interviews with national (n = 122) and sub-national (107) level Network members and key stakeholders were conducted. Several iterative rounds of interviews were conducted in each country, typically at least six months apart, to capture (a) changes in how the Network was operating, (ii) views pertaining to Network activities at the time of interview, and (iii) follow-up on emerging findings from the previous round. The participants were recruited purposively by identifying MOH and partner organizations involved in QCN who could provide rich information about the Network (Table 2).

Table 2.  Qualitative interviews and health facility observations completed, by time, in each country.

| Case-study Country | Data collection dates | National interviewee (n) | Sub-national Interviewee (n) | Facility Observation (n) |
|---|---|---|---|---|
| Bangladesh | 1 (Oct 2019 –Mar 2020) | 13 | 7 | 3 |
| | 2 (Oct 2020 –Jan 2021) | 14 | 11 | 0 |
| | 3 (May 2021 –Sep 2021) | 10 | 12 | 4 |
| | 4 (Jan 2022 –Mar 2022) | 8 | 0 | 0 |
| Ethiopia | 1 (Jan 2021– Mar 2021) | 8 | 11 | 4 |
| | 2 (Nov 2021 –Dec 2021) | 10 | 11 | 3 |
| Malawi | 1 (Oct 2019 –Mar 2020) | 7 | 12 | 4 |
| | 2 (Nov 2020 –Jan 2021) | 10 | 7 | 4 |
| | 3 (Aug 2021 –Nov 2021) | 9 | 7 | 4 |
| | 4 (Mar 2022) | 2 | 3 | 0 |
| Uganda | 1 (Nov 2020 –Mar 2021) | 7 | 13 | 4 |
| | 2 (Jun 2021 –Sep 2021) | 12 | 8 | 4 |
| | 3 (Feb 2022 –Mar 2022) | 10 | 5 | 4 |

Concurrently, semi-structured interviews were also conducted with QCN global actors (n = 7 in Mar-2021 and n = 14 during Nov-2021–Feb-2022). The number of interviews at each setting was based on having sufficient information saturation to answer our research questions. These interviews explored views on attributes of QCN and its operational strategy and performance that might affect the sustainability of QCN, among other things (S2 Text).

**Non-participant observations.** Second, non-participant observations were conducted. In QCN health facilities, these were conducted via visits to two well and two least performing QCN health facilities in each case study country in two to three iterative rounds (Table 2). Well and least performing QCN health facilities were purposively selected through discussion with key stakeholders and review of facility-level maternal and newborn health outcome and other quality of care data (e.g., those used in national schemes). During these facility observations, structured templates were used to capture key processes relevant to the focus of the Network in each country, as well as unstructured notes. In addition, non-participant observations of key national-level and district level meetings were conducted during which processes and priority discussion topics were captured through unstructured notes. These meetings were usually organized by national level actors such as MOH and the schedule and purpose of the meeting was communicated by the host or during partner interviews. Finally, one global level QCN meeting was observed during the study period.

**Analysis.** We performed a thematic analysis of the qualitative interviews and observations. A framework approach [32] was used to analyse the data based on a priori themes around six critical actions summarised by Wickremasinghe and others to define the actions that actors at different levels can take to help sustain innovations (Table 3). We developed a matrix based on the themes, and codes that fall under each theme were assigned (S1 Table). All the co-authors reviewed and approved the matrix. Then the data was charted into the matrix for each country including the quotes that represent the summary data. We analysed and interpretated the data for each country first and after receiving feedback from each country data lead, the results were further analysed and interpreted, identifying similarities and differences across countries and results were presented using the six sustainability actions. We defined community as patients, clients of the health service, families or members of local community who have stake in the health service provision.

**Table 3. Six critical actions to help sustain innovations [10].**

| # | Critical action | Rationale |
|---|---|---|
| 1 | Planning opportunities for reflection and adaptation from the outset | Building in the expectation that there will be a need to continuously learn, reflect and adapt processes can help innovations be fit for purpose in the real world |
| 2 | Strong government ownership | Enabling government leadership in planning, inception and implementation strengthens the potential for commitment to, and responsibility for, innovations in the longer term |
| 3 | Motivating micro-level actors | Consideration of the needs and preferences of local-level implementers is essential for most innovations |
| 4 | Institutionalizing the innovation within the health system | Integration of processes (eg supervision, supply chain, data) within existing systems promotes ownership, reduces duplication, improves efficiency |
| 5 | Managing financial uncertainties | Seeking sustained financial commitment from government, e.g. adding innovation costs to strategic plans and budgets, works alongside institutionalization and can help to minimise the impact of system shocks, e.g. a change in government. |
| 6 | Fostering community ownership | Community groups can be important advocates for the continuation of innovations and hold leaders to account |

### Ethics

All data collection was conducted after obtaining written consent, including separate consent for tape recording. Patients' privacy was respected during hospital observations. Our study didn't include minors as study participants. All data is confidential and anonymised. Ethical approval was obtained from the Research Ethics Committee at University College London (3433/003); institutional review boards in Bangladesh, BADAS Ethical Review Committee (ref: BADAS-ERC/EC/19/00274), Ethiopian Public Health Institute Institutional Review Board (ref: EPHI-IRB-240-2020), National Health Sciences Research Committee in Malawi (ref: 19/03/2264) and Uganda Makerere University School of Public Health- Higher degrees Research Ethics Committee in Uganda (ref: Protocol 869)

## Results

Results are synthesized across the experience of the learning districts and health system of four QCN countries. We draw on the evidence described in Table 2, in addition to the interviews and observations with global level actors to identify whether each action was present and how it influenced the potential for Network sustainability at the scale it had been implemented at during this investigation. To give a snapshot of experience by country, we also present a high-level summary of these actions by country (Table 4). Overall, the evidence from Bangladesh suggested that all sustainability actions were present during QCN implementation to a certain degree. Other countries experienced more limited engagement across the set of actions, especially apparent around managing financial uncertainty and fostering community engagement.

## 1. Planning opportunities for reflection and adaptation

All respondent types interviewed reported that opportunities for planning, reflection and adaptation were embedded in the Network approach at the global, national, and sub-national levels, although some vulnerability was described in Malawi and Uganda.

At the global level, between countries, respondents recalled the importance of holding repeat, joint international meetings with global partners, held in Malawi in 2017, Tanzania in 2018, and Ethiopia in 2019. These meetings promoted the importance of country engagement with the Network and encouraged learning. A respondent in Bangladesh noted:

> "But I was in that [QCN] meeting along with the government . . . . the ministry agreed, and the team participated in that Malawi workshop. . . . . we had highest policy level commitment to participant in the QCN network" (Implementing Partner- National-Bangladesh Round 1).

**Table 4. Status of the sustainability actions in the four QCN countries.**

| Sustainability actions | Bangladesh | Ethiopia | Malawi | Uganda |
|---|---|---|---|---|
| 1. Planning opportunities for reflection and adaptation | 🟩 Green | 🟩 Green | 🟧 Yellow | 🟧 Yellow |
| 2. Government ownership with a plan for a phased transition | 🟧 Yellow | 🟧 Yellow | 🟧 Yellow | 🟧 Yellow |
| 3. Motivating micro-level actors | 🟩 Green | 🟩 Green | 🟩 Green | 🟩 Green |
| 4. Institutionalizing the innovation within the health system, | 🟧 Yellow | 🟧 Yellow | 🟧 Yellow | 🟧 Yellow |
| 5. Managing financial Uncertainties | 🟧 Yellow | 🟥 Red | 🟥 Red | 🟥 Red |
| 6. Fostering community engagement | 🟧 Yellow | 🟥 Red | 🟧 Yellow | 🟥 Red |

\***Green** represents the weight of evidence suggest the presence of the action on multiple accounts, if not all. **Yellow** represents that evidence indicates the action to be present to some degree, but with some vulnerability or weakness. **Red** represents there is no evidence in the data to indicate the action exists.

However, some respondents commented that there was limited follow-up and support from the global actors to see if the learning at the global level was adopted at the national level.

At the national level, respondents acknowledged opportunities for reflection and adaptation from the outset in the form of joint consultative meetings and joint assessments. During these meetings, activities were planned, learning sites selected, and then partner organisations contributions discussed and coordinated. This type of national level engagement was particularly strongly reported in Ethiopia, including the MOH and partner organizations organising a joint quality summit in the country.

In all countries there was also evidence that the generic quality of care standards from the WHO Quality of Care framework were adapted to meet the needs of government quality management directorates. A respondent capitalized on the importance of contextualizing interventions at the country level as follows:

*". . .economically we are different, the setups of the government are different. For instance, we take Malawi, and we compare it with South Africa its [implementation] will be totally different but the standards will be the same."* (Government-Local case 1-Malawi round 2)

Finally, at sub-national levels in all countries, the restriction of implementation to a small number of learning health facilities, with the intention to foster learning for future scale-up, automatically implied built in opportunity for reflection and adaptation. These learning sites also had opportunities for reflection during the learning forums where health facilities with better performance in QI work shared their experience. However, linkages between reflective learning at national and sub-national levels did not always lead to adaptation in practice, for example in Uganda and Malawi where QCN structures at the sub-national and local level were reported to be less strong respectively.

*"At the district level, they have known their part in the Network but at facility level we don't really mention the Network. We mention it during training, but they are not that conscious about it, although they know that there are facilities within the district that are also implementing and that they need learn and share and thus should hold meetings every quarter to come together and learn from each other. The importance of the Network at the district level is not so high, it is more at the national level."* (Government-National-Uganda round 2)

## 2. Strong government ownership with a plan for a phased transition

All countries demonstrated strong ownership of the QCN, at least in terms of political and normative commitments. However, none of the countries had a plan for transition when QCN partners had completed their contract of implementation.

At the global level, there was a push for country governments to take ownership of their respective QI activity. WHO provided technical support, developing guidelines and frameworks such as the LALA (leadership, action, learning and accountability) framework which facilitated the implementation and monitoring of the Network activity at national and sub-national levels. WHO's approach of leadership was also appreciated as being non-prescriptive, actively seeking buy-in and ownership from partners and country governments.

At the national level, the MOH of each country took ownership of the QCN initiative and enlisted partner support. In Ethiopia, QCN was reported as the Ministry's flagship program, creating technical committees and organizing partners' efforts. Similarly, in Bangladesh a government academic institution, National Institute of Preventive and Social Medicine (NIPSOM) supported MoH in the implementation of the Network, together with other partners.

*"The Government has many initiatives especially in the context of quality of care. It's basically a government program."* (Implementing Partner-National-Bangladesh round 1).

However, a vulnerability that was reported across countries was the fact that the national MOH quality directorates worked in isolation; it was thought that better integration of quality across directorates could further strengthen ownership of the Network.

*"I think one other problem we have in higher offices in the ministry is that programs are working in isolation. And we know worldwide that we cannot achieve quality, or we cannot make quality improvement if we try to work as individuals. So, the departments need to come together and be seen of the ground together and move forward"* (Health facility worker-Local case 4-Malawi round 3)

Between countries, government ownership was not uniform at the sub-national level. In Malawi, the structure that was established at the national level went to the lower-level health system, down to the community. In Bangladesh, the Civil Surgeons took leadership of the QI activities at sub-national level. But in Ethiopia and Uganda government ownership was relatively weaker at the sub national level. In Uganda, the system didn't cascade down to the lower level of the health system and in Ethiopia a lack of commitment was observed from the regional health system. A respondent from Ethiopia commented:

*"We have no role in the Network so there cannot be conflict of interest. We do mentorship & coaching at three hospitals. Other than that, the structure is not stretched down. At the office level, we are not required to provide support. . . . . . . to be frank the plan is not ours; it is MOH's plan."* (Government-Local-Ethiopia Round 1)

Although the MOH of the respective countries took ownership of the Network, and activities took place within existing structures, a particular vulnerability was that implementation was usually facilitated by the implementing partners through individual projects.

*". . . regionalized support like UNICEF is already in certain districts, so they have been supporting that work in their districts. That's how it's been working and then Government sort of takes the middle piece where if there is capacity building, they support that, although other partners have also done their part in capacity building and trainings within their budgets."* (Government-National-Uganda round 2)

While the strong ownership and coordination at national level was positive, the more fragmented ownership sub-nationally, and the approach of partners implementing activities on a project basis, limited the opportunity for a phased transition of responsibilities in all countries. Some partners did not have a vision for long term engagement, beyond their current funding, and sub-national leaders did not feel confident that they would have the resources to implement without partner support.

*". . .you know some other partners just come and then disappear. So sometime other partners are inactive, and some partners will come and say, I think our funding has finished. And when their funding has finished they just disappear, and they even don't say anything and this has been a problem"* (Government-Local case 2-Malawi round 2)

### 3. Motivating micro-level actors

All four countries employed various mechanisms to motivate the healthcare workers and those supporting the work of health facilities at a grass root level in relation to the QCN work. An incentive mechanism in the form of small funding or grants for health facilities was reported in Bangladesh, Ethiopia, and Uganda as part of the QCN intervention. This incentive approach was appreciated by the respondents because it created an enabling environment for the health workers to be innovative in identifying, prioritizing, and solving problems within their health facility. The incentive was also given to their health facility as a form of reward for the best performer in quality service provision.

> *"They [facility workers] come [to a fair] and participate in a competition. Whose performance is the best according to the report? An award is given according to the [facility] performance. It is given facility wise and inter-district wise."* (Government-Local case 1-Bangladesh round 1)

In addition, several respondents confirmed that the knowledge and skills gained through the extensive training linked to the Network activities further fostered motivation.

> *"Without having knowledge, there is no motivation to do the work. Now when they realized that they could do better, now they do the work with more enthusiasm and do the work with more quality."* (Government-Local case 1-Bangladesh round 1).

> *"The activities[training] are nice because it fills the skill gaps. As you know even though most of our workers have theoretical knowledge they lack skills. . . ..In the process of filling the skill gaps indicators are presented, detail technical works are also included. Because of this, I am interested in the activities. These are technical duties that help professionals to follow every step to provide health services."* (Government-Local-Ethiopia round 1)

Nonetheless, despite the positive comments on QCN actions to motivate micro-level actors, two areas of concern were broadly noted. First that while such incentives were observed to positively motivate micro-level actors during this phase of QCN, the use of financial incentives for individuals might not be sustainable in the longer term or if QCN activities were scaled up beyond the current learning areas. And second, if deficiencies in health facility structural quality persisted into the future, or if career progression for health workers was limited, then the QCN actions to motivate the workforce would be weakened.

> *". . .there are a lot of demotivators yah? Maybe career paths. Frustrations also come with small issues like infrastructure in which the staff are working in."* (Government-National-Malari round 2)

### 4. Institutionalizing the innovation within the health system

Implementers in all four countries were keen to work within the existing health system, to avoid creating parallel systems, and to enable the physical environment for QI through investment in existing infrastructure, job aids and guidelines. In each country, Network activities were located within a designated government unit or department that was responsible for health care quality. For example, in Ethiopia, QCN had a designated person at each level of the health system, and activities were coordinated as part of the national plan, with some variation at sub-national levels. In Malawian hospitals, Network activities were integrated in Quality Management Units, working through pre-existing Quality Improvement Support Teams.

However, the institutionalization of QI in Malawi was not perceived to be adequate and respondents suggested to have QI as part of the tertiary level education, so that health workers would have adequate knowledge and understanding of QI when they joined the workforce.

*"My best bet would be to have as many officers, as many frontline workers playing in quality . . .We should be really thinking about. . . if graduates are coming straight from college, they should already know that quality is built in every clinical program and that it's not something that is separate, but it is part of that clinical training. So, the training in MNH, then QI is part of it because quality is eventually what we need. . . that's how we serve a customer."* (Implementing partner-National-Malawi round 3)

Learning forums and training were thought to play an important role in institutionalizing QI in the health system. The learning forums allowed transfer of knowledge and skill within and across health facilities and these were shared by health workers and managers with their colleagues and remained in the health system. As was the advocacy work that partners carried out to raise awareness about the initiative. An example was reported from Ethiopia of a region that had started to prepare a quality improvement bulletin to give more voice to the Network idea.

However, four vulnerabilities emerged that limited institutionalisation efforts. First, the consequences of losing partner support at the end of their funded project period was described as a problem that weakened the Network as MOH struggled to fill the gap and maintain momentum.

*". . .what scares us most is the question 'If the partners left, would the initiative continue*?'. *They are very supportive of QI projects. As I said if you go to the district level and observe you may observe many QI projects. This is due to the partner organizations. . . . Sometimes I wonder if the program only lasts as long as those partners exist. Perhaps if they left, I am not sure about the continuity. But for now, it is good."* (Government-National- Ethiopia round 1)

Second, respondents mentioned that partner priorities did not always perfectly align with the real-world needs in the country, especially at sub-national levels where de-centralised decision making was needed. As reported in Uganda, multiple partners invested on the same activity when it was known that it was not a priority for the district. A respondent from Malawi also described existing misalignment between partner and government priorities as follow:

*"the challenge with our partners when they are coming into they have their own objectives to achieve that may be line with what we want but they are coming in the name of quality but not on the specifics that we are targeting so thus what I can say over that one"* (Government-Local case 1- Malawi round 2)

Third, some respondents reported fragmentation of implementation according to the presence of different implementing partners who had different organizational missions and vision. This was emphasised by respondents from Bangladesh, where the implementing partners divided the implementing areas among themselves, but activities carried out according to their own pace, with different level of intensity.

And finally, the COVID 19 pandemic shifted both emphasis and resources away from the quality improvement activities and tested the strength and depth of institutionalization of the Network activities within the health system. A respondent from national implementing

partner in Ethiopia reported that because of the COVID-19 outbreak, their organization had to close all its program including QCN and transferred their budget to COVID-19 response.

## 5. Managing financial uncertainties

Initial Network initiatives in all countries were heavily supported by implementing organizations through external funding. As seen at the global level, the funding for QCN came primarily from the Bill and Melinda Gates foundation (BMGF) and USAID; the contribution of WHO through its staff time was also noted. At national level, however, some progress of financial commitment from government was observed, particularly for coordination efforts, though less so for implementation, which was mostly still dependent on partner organisation support. In Ethiopia and Uganda, there was some evidence of government financial support or budget allocated to QI. And in Bangladesh, several respondents noted the government's longstanding commitment to achieving universal health coverage, consistent with the goals of the Network. Here, where QCN was observed to be particularly well assimilated in government plans, it was impossible to see Network activities separately from government QI actions, creating a strong belief that the government would manage financial uncertainties, as exemplified by a respondent from a partner organization:

*"It's a project that you are talking about, but we are not concerned about the time of QCN project because the quality improvement initiative that we are doing is part of the government plan, there is nothing with that QCN project. Even we don't use this term QCN, so this is part of our sector programme. This is the way we are supporting; we are taking it forward as part of their operational plan and sector plan. And now they have developed the quality strategy and now we'll develop the action plan, and they will go beyond 2022. . .."* (Implementing partner-National- Bangladesh round 4)

But other countries expressed concern about the continuity of QCN efforts in the absence of external funding. Although Ethiopia did try to manage interruption of funding when an individual support partner phased out by committing budget to QCN activities, this effort of the government was jeopardised by external shocks such as COVID 19.

Similarly in Malawi, a respondent commented about the fate of QCN in the absence of external funding:

*"But I find the issue to do with financing more of a cause for us to fail. This is because look at all the components of the health system and I find. . . well. . . I was trying at this particular time to think about the investments that have happened for example in Kasungu, as a learning district. How much did government commit to the goal that we reduce the maternal mortality rate by fifty per cent in the implementing (of the project) in the nation and districts by 2022? If we are to be honest, success of every implementing district was dependant on the kind of and the flexibility of partners that are in the district."* (Implementing partner-National-Malawi round 3).

## 6. Fostering community ownership and acceptance

All four countries had a system for community engagement, but it was seldomly used for the purpose of the QCN except in Bangladesh and Malawi. Similarly, there was little emphasis on community engagement in relation to QCN at the global level, despite community empowerment being a central pillar of WHO's theory of change for QCN. Community engagement was particularly strong in Bangladesh, perhaps reflecting the relatively strong health system there

prior to QCN implementation. Community leaders supported QI work in hospitals and took part in monthly coordination meetings organized by the district leadership; members of the public participated in QI activities through volunteer groups and clubs; partner organizations established suggestion boxes, help desks, citizen charters and community score cards to promote community voice; and government created platforms for community meetings to advocate for quality improvement. All these initiatives were present prior to QCN but had been aligned and adapted for the same purpose. However, few said that the community engagement part is still a working progress and yet to be designed and implemented as part of the QCN work by their implementing organization.

*". . ..WHO has released a stakeholder and community engagement module. And from that module we have some ideas and some guidelines; how we should communicate with the community for this quality improvement. Right now we are in a process of developing the Bangladesh based context module based on that WHO module. . ."* (Implementing partner-National-Bangladesh round 4)

In Malawi, community engagement was added as the ninth standard in the MNH QoC standards. A formal structure to link the community members with service providers in health facilities was established, called Health Centre Advisory Committee. This committee was responsible not only for promoting accountability but mobilized resources for the QI initiatives. Village Health Committees and Village Development Committees also played a key role in mobilizing resources.

*"there is a feedback mechanism where like if clients are not satisfied with the services that they are receiving or maybe a certain injustice has happened they do complain to the ombudsman and their issues get resolved. The hospital ombudsman also conducts some exit interviews where they check the satisfaction level of the quality of services that are being offered at the facility. So at the end of the month, the HO produces an exit report on how many clients they interviewed, how many were not satisfied with the services and the reasons for lack of satisfaction and others things. . ."* (Government-National-Malawi round 2)

However, not all agreed on the extent of community engagement in Malawi.

*". . ..it was found that standard nine(community engagement) is the one that is not being implemented in almost all the districts. There is a big challenge on the one that talks about community and accountability. . . so issues of score card is not done. . . so it's almost cut across."* (Government-Local case 3-Malawi round 3)

Despite the existence of strong community engagement structures in Uganda and in Ethiopia (for example through the Health Extension Programme in Ethiopia), community involvement did not emerge as a strong component of Network activity. One participant reflected that this might have been an oversight that could subsequently be addressed.

*"Then when we come to the stakeholders and community engagement, we are not doing so well, UNICEF has done some work to this business of community engagement using Village Health Teams [the lowest point of Uganda's health system]. But there is a gap of not engaging the health unit management committees [HUMCs] which bridge the community with facilities."* (Implementing partner-National-Uganda round 1)

## Discussion

Our analysis examined the presence of six critical actions to support sustainability of QCN in the limited number of implementation areas in four Network countries. Institutionalization of the innovation with the health system and motivating micro-level actors were found in all countries, while recognising that some vulnerability existed. There was also some evidence of actions taken to plan opportunities for reflection and adaptation from the outset and to support strong government ownership. However, these actions were stronger at national than subnational level. Two actions were largely absent and weakened confidence in future sustainability: managing financial uncertainties and fostering community ownership.

Institutionalization of QCN within existing systems was strong in all four countries, and particularly to the extent that QCN in Bangladesh and Ethiopia was recognized as part of the governments' QI initiative, not as a separate entity. The alignment of goals of QCN with country priorities and their desire to improve the health of mothers and newborn in all four countries positioned QCN as a favoured intervention. Building and sustaining institutional capability including the local capability was reported as a means to sustain a scale-up of an innovation [9, 33]. However, we also witnessed that institutionalization could be affected in the presence of financial uncertainty as in Ethiopia, poor harmonization of effort among implementing partners as in Bangladesh, and suboptimal alignment of country needs with implementing partners objectives at sub-national level as in Uganda and Malawi.

All countries took essential steps in motivating micro-level actors, although the sub-optimal environment in which these actors worked sometimes operated against the motivating actions as reported elsewhere [34]. But QCN was regarded as a beneficial initiative for staff. The training and knowledge and skill sharing sessions were most appreciated sources of motivation together with the financial incentives given to health facilities based on their performance in QI. In many low-and middle-income countries there is an insufficient number of health workforce, including in the case study countries [35]; actions to motivate health workers are important for retention in the health system [11, 14]. Training was reported as a source of motivation for health workers in previous studies [36, 37] as was improving the environment they operated in [34, 36].

Opportunities for reflection and adaptation of QCN were embedded in the design with repeat learning forums at all levels. The fact that governments took the initiative to engage in conversations before embarking on QCN activity in all countries created a strong starting platform for country implementation. In addition, country commitment to global initiatives such as the SDGs created a fertile ground for QCN to act as a catalyst to achieve these global commitments. The learning forums and meetings that happened at the global, national, and subnational level set the stage for country adaptation of QCN, crucial for accommodation of country specific contexts [11]. However, accountability for implementing learning was not optimal everywhere because of weak systems and realising opportunities for learning often relied on external support [38]. Further, more time, effort and engagement were needed at the local level to secure leadership commitment and resource.

There was strong government ownership of the QI initiative in all countries [39]. From the start, QCN was not rigidly prescribed by the global actors unlike many donor-funded interventions. But two areas of vulnerability included that government ownership did not extend to all levels of the health system [14, 40]; and while there was confidence that QCN would continue to be a government priority going forward, none of the countries had a plan for phased transition from partner support to full government implementation. The lack of a plan for phased transition had already affected the Ethiopia program as some of the implementing partners had already completed their contracted support.

Of the two actions observed to be less present, financial uncertainty limited the ability of the four countries to move forward in the absence of continuous support and none of the countries had a financial sustainability plan. This limitation necessarily challenges the question of the strength of ownership by country governments [10, 13], but also challenges global partners to ensure that achieving financial security was central to the design. In Ethiopia and Uganda there were some attempts by the government to fill gaps in funding during the QCN implementation period. However, we didn't identify any plan laid out to manage the financial uncertainties, except the strong optimism from respondents in Bangladesh.

Finally, engaging the community as a sustainability action received relatively little attention, except in Bangladesh and Malawi where there was some evidence of community engagement to the extent of mobilizing domestic resources for the initiative. However, both Ethiopia and Uganda made little effort to utilize their already well-established community health system [41]. Other studies acknowledged the benefit of engaging the community in such innovative interventions to ensure community acceptance and its sustainability [10, 11, 14, 42]. Defining community engagement or ownership in the context of QCN may be crucial to maximize gain from the community engagement process, especially in the countries where their roles in QCN was not yet defined [43].

## Strengths and limitations of this study

This analysis triangulated data from key partners at the global, national, and sub-national level in the four case study countries that improved the credibility of our findings. Important insights were observed about actions taken that promoted the sustainability of QCN. But the evaluation could only make inference in the context of implementation in a relatively small number of implementation districts, and over a relatively short period of implementation; it did not attempt to engage with sustainability at scale. Further, while national level participant meetings were observed, meetings at the district level were not included in the original plan: it is possible that this limited our understanding at the implementation level however, given the depth of information from individual interviews, it is unlikely to change our findings. The framework of six sustainability actions was a useful tool with which to examine whether and how the innovation could be sustained for the longer term, but some co-dependence was observed between actions such that, for example, positive remarks about government ownership and institutionalisation were made vulnerable by financial uncertainty.

## Conclusion

The framework of six critical actions to promote sustainability was useful in revealing where progress was made and what more could be done to sustain improvements in MNH outcomes and quality of care. The innovation was observed to be relatively top-down, with the drive being strongest at global and national levels but with much work–and time—needed to embed QCN linked activities at the sub-national level. Crucially, it was revealed that the absence of deliberate action to address financial uncertainty was an obstacle to the sustainability of QCN. Nevertheless, the strong progress made to institutionalize some characteristics of QCN in existing government systems should be supported to avoid any stalling of progress.

## Supporting information

**S1 Text. PLOS global public health QCN Evaluation collection 2-page summary.** (DOCX)

**S2 Text. PLOS global public health QCN papers common methods section.**
(DOCX)

**S1 Table. Data coding and analysis matrix.**
(DOCX)

## Acknowledgments

We would like to thank all study respondents for their invaluable participation in the overall QCN evaluation. We also appreciate the participation of the QCN Evaluation Group, made of: Fatama Khatun, Abdul Kuddus, Kishwar Azad (BADAS-PCP Bangladesh), Gladson Monjeza, Racheal Magaleta, Zabvuta Moffolo, Charles Makwenda (Parent and Child Health Initiative, Malawi), Mary Kinney, Fidele Mukinda (independent researchers, South Africa), Will Payne, Jeremy Shiffman (Johns Hopkins University, USA), Hilda Namakula, Agnes Kyamulabi (Makerere University, Uganda) Anene Tesfa, Theodros Getachew, Geremew Gonfa (Ethiopia Public Health Institute, Ethiopia).

## Author Contributions

**Conceptualization:** Seblewengel Lemma, Tanya Marchant.

**Data curation:** Seblewengel Lemma, Callie Daniels-Howell, Asebe Amenu Tufa, Mithun Sarker, Kohenour Akter, Catherine Nakidde, Albert Dube.

**Formal analysis:** Seblewengel Lemma, Asebe Amenu Tufa, Mithun Sarker, Kohenour Akter, Catherine Nakidde, Albert Dube.

**Funding acquisition:** Gloria Seruwagi, Mike English, Yusra Ribhi Shawar, Tim Colbourn, Tanya Marchant.

**Investigation:** Seblewengel Lemma, Callie Daniels-Howell, Asebe Amenu Tufa, Mithun Sarker, Kohenour Akter, Catherine Nakidde, Gloria Seruwagi, Albert Dube, Kondwani Mwandira, Desalegn Bekele Taye, Mike English, Yusra Ribhi Shawar, Kasonde Mwaba, Nehla Djellouli, Tim Colbourn, Tanya Marchant.

**Methodology:** Callie Daniels-Howell, Mithun Sarker, Kohenour Akter, Catherine Nakidde, Gloria Seruwagi, Albert Dube, Kondwani Mwandira, Mike English, Yusra Ribhi Shawar, Kasonde Mwaba, Nehla Djellouli, Tim Colbourn, Tanya Marchant.

**Project administration:** Seblewengel Lemma, Callie Daniels-Howell, Kasonde Mwaba, Nehla Djellouli, Tim Colbourn, Tanya Marchant.

**Resources:** Callie Daniels-Howell, Kasonde Mwaba, Nehla Djellouli, Tim Colbourn.

**Software:** Nehla Djellouli.

**Supervision:** Kohenour Akter, Kondwani Mwandira, Mike English, Yusra Ribhi Shawar, Kasonde Mwaba, Nehla Djellouli, Tim Colbourn, Tanya Marchant.

**Validation:** Seblewengel Lemma, Callie Daniels-Howell, Mithun Sarker, Kohenour Akter, Catherine Nakidde, Gloria Seruwagi, Albert Dube, Kondwani Mwandira, Desalegn Bekele Taye, Mike English, Yusra Ribhi Shawar, Kasonde Mwaba, Nehla Djellouli, Tim Colbourn, Tanya Marchant.

**Visualization:** Seblewengel Lemma.

**Writing – original draft:** Seblewengel Lemma.

**Writing – review & editing:** Seblewengel Lemma, Callie Daniels-Howell, Asebe Amenu Tufa, Mithun Sarker, Kohenour Akter, Catherine Nakidde, Gloria Seruwagi, Albert Dube, Kondwani Mwandira, Desalegn Bekele Taye, Mike English, Yusra Ribhi Shawar, Kasonde Mwaba, Nehla Djellouli, Tim Colbourn, Tanya Marchant.

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
