## [Decision Letter · Decision Letter 0]

8 Mar 2023

PGPH-D-22-01888

Opportunities to sustain a multi-country quality of care network: lessons on the actions of four countries Bangladesh, Ethiopia, Malawi, and Uganda

Dear Dr. Abreham,

Thank you for submitting your manuscript to PLOS Global Public Health. After careful consideration, we feel that it has merit but does not fully meet PLOS Global Public Health’s publication criteria as it currently stands. Therefore, we invite you to submit a revised version of the manuscript that addresses the points raised during the review process.

Please note that we have only been able to secure a single reviewer to assess your manuscript. We are issuing a decision on your manuscript at this point to prevent further delays in the evaluation of your manuscript. Please be aware that the editor who handles your revised manuscript might find it necessary to invite additional reviewers to assess this work once the revised manuscript is submitted. However, we will aim to proceed on the basis of this single review if possible. 

We look forward to receiving your revised manuscript.

Kind regards,

Vanessa Carels

Staff Editor

Journal Requirements:

1. Please provide additional details regarding participant consent. In the ethics statement, please ensure that you have specified what type you obtained (for instance, written or verbal, and if verbal, how it was documented and witnessed). If your study included minors, state whether you obtained consent from parents or guardians. If the need for consent was waived by the ethics committee, please include this information."

2. Please ensure that Funding Information and Financial Disclosure Statement are matched.

Additional Editor Comments (if provided):

Reviewers' comments:

Reviewer's Responses to Questions

**Comments to the Author**

1. Does this manuscript meet PLOS Global Public Health’s publication criteria? Is the manuscript technically sound, and do the data support the conclusions? The manuscript must describe methodologically and ethically rigorous research with conclusions that are appropriately drawn based on the data presented.

Reviewer #1: Partly

2. Has the statistical analysis been performed appropriately and rigorously?

Reviewer #1: N/A

3. Have the authors made all data underlying the findings in their manuscript fully available (please refer to the Data Availability Statement at the start of the manuscript PDF file)?

Reviewer #1: No

4. Is the manuscript presented in an intelligible fashion and written in standard English?

Reviewer #1: Yes

5. Review Comments to the Author

Reviewer #1: Why were there variations in the data collection dates and the number of times data was collected for the different countries ? there should be an explanation

What is the rationale behind the number of participants interviewed or the number of transcripts analyzed in the member countries

The 1st sustainability action states "At the global level, between countries, respondents recalled the importance of holding repeat, joint international meetings with global partners" but according to table 2 of the methods supplement, global meeting minutes were not reviewed, is this because they don't exist or is there another specific reason? In Uganda also no minutes were reviewed, why?

There were no observations made at local/district level meetings, would it not have been critical to this study to undertake those observations at the implementation level?

Regarding sustainability action 3,

It says all four countries employed various mechanisms but then only discusses one i.e small grants for health facilities what are the other types of motivating factors used and were such incentives applied in all countries, is there a quote to support that? this makes one skeptical about the all green scoring in table four for this particular theme, wouldn't you say?

There were missed opportunities to ask WHY and go deeper into understanding the root causes of such findings.

How do the findings inform future program design and possibly evaluation, for instance seeing that Learning is an essential pillar of the LALA framework but this study indicated that some countries were doing good in one category and others in other categories, why did they fail to learn from one another?

With the exception of Ethiopia, all countries had at least three rounds of interviews, were there any changes in the responded across time, how is that reflected in this study?

6. PLOS authors have the option to publish the peer review history of their article (what does this mean?). If published, this will include your full peer review and any attached files.

**Do you want your identity to be public for this peer review?** For information about this choice, including consent withdrawal, please see our Privacy Policy.

Reviewer #1: No

---

## [Decision Letter · Decision Letter 1]

12 Jul 2023

PGPH-D-22-01888R1

Opportunities to sustain a multi-country quality of care network: lessons on the actions of four countries Bangladesh, Ethiopia, Malawi, and Uganda

Dear Dr. Abreham,

Thank you for submitting your manuscript to PLOS Global Public Health. After careful consideration, we feel that it has merit but does not fully meet PLOS Global Public Health’s publication criteria as it currently stands. Therefore, we invite you to submit a revised version of the manuscript that addresses the points raised during the review process.

Unfortunately, the original reviewer was not available to assess your revisions. We have therefore invited another reviewer, who has raised a few minor concerns. Please see the full report below and attached.

We look forward to receiving your revised manuscript.

Kind regards,

Dario Ummarino, PhD

Staff Editor

Journal Requirements:

1. Please provide additional details regarding participant consent. In the ethics statement, please ensure that you have specified what type you obtained (for instance, written or verbal, and if verbal, how it was documented and witnessed). If your study included minors, state whether you obtained consent from parents or guardians. If the need for consent was waived by the ethics committee, please include this information.

Additional Editor Comments (if provided):

Reviewers' comments:

Reviewer's Responses to Questions

**Comments to the Author**

1. If the authors have adequately addressed your comments raised in a previous round of review and you feel that this manuscript is now acceptable for publication, you may indicate that here to bypass the “Comments to the Author” section, enter your conflict of interest statement in the “Confidential to Editor” section, and submit your "Accept" recommendation.

Reviewer #2: (No Response)

2. Does this manuscript meet PLOS Global Public Health’s publication criteria? Is the manuscript technically sound, and do the data support the conclusions? The manuscript must describe methodologically and ethically rigorous research with conclusions that are appropriately drawn based on the data presented.

Reviewer #2: Yes

3. Has the statistical analysis been performed appropriately and rigorously?

Reviewer #2: N/A

4. Have the authors made all data underlying the findings in their manuscript fully available (please refer to the Data Availability Statement at the start of the manuscript PDF file)?

Reviewer #2: No

5. Is the manuscript presented in an intelligible fashion and written in standard English?

Reviewer #2: Yes

6. Review Comments to the Author

Reviewer #2: Dear Authors,

I would like to congratulate you for an interesting and well-written manuscript.

I have very minor comments or edits I have suggested which hope you will consider for the final version of your manuscript.

I have no additional comments,

Yours sincerely.

7. PLOS authors have the option to publish the peer review history of their article (what does this mean?). If published, this will include your full peer review and any attached files.

**Do you want your identity to be public for this peer review?** For information about this choice, including consent withdrawal, please see our Privacy Policy.

Reviewer #2: **Yes: **Mohsin Sidat

---

## [Editor Report · Decision Letter 2]

11 Aug 2023

Opportunities to sustain a multi-country quality of care network: lessons on the actions of four countries Bangladesh, Ethiopia, Malawi, and Uganda

PGPH-D-22-01888R2

Dear Dr. Abreham,

We are pleased to inform you that your manuscript 'Opportunities to sustain a multi-country quality of care network: lessons on the actions of four countries Bangladesh, Ethiopia, Malawi, and Uganda' has been provisionally accepted for publication in PLOS Global Public Health.

Best regards,

Julia Robinson

Executive Editor